# Improving the Learning Capability of Small-size Image Restoration Network by Deep Fourier Shifting

**Man Zhou**

Aerospace Information Research Institute, Chinese Academy of Sciences
University of Science and Technology of China

## Abstract

State-of-the-art image restoration methods currently face challenges in terms of computational requirements and performance, making them impractical for deployment on edge devices such as phones and resource-limited devices. As a result, there is a need to develop alternative solutions with efficient designs that can achieve comparable performance to transformer or large-kernel methods. This motivates our research to explore techniques for improving the capability of small-size image restoration standing on the success secret of large receptive filed.

Targeting at expanding receptive filed, spatial-shift operator tailored for efficient spatial communication and has achieved remarkable advances in high-level image classification tasks, like $S^2$-MLP [1] and ShiftVit [2]. However, its potential has rarely been explored in low-level image restoration tasks. The underlying reason behind this obstacle is that image restoration is sensitive to the spatial shift that occurs due to severe region-aware information loss, which exhibits a different behavior from high-level tasks. To address this challenge and unleash the potential of spatial shift for image restoration, we propose an information-lossless shifting operator, i.e., Deep Fourier Shifting, that is customized for image restoration. To develop our proposed operator, we first revisit the principle of shift operator and apply it to the Fourier domain, where the shift operator can be modeled in an information-lossless Fourier cycling manner. Inspired by Fourier cycling, we design two variants of Deep Fourier Shifting, namely the amplitude-phase variant and the real-imaginary variant. These variants are generic operators that can be directly plugged into existing image restoration networks as a drop-in replacement for the standard convolution unit, consuming fewer parameters. Extensive experiments across multiple low-level tasks including image denoising, low-light image enhancement, guided image super-resolution, and image de-blurring demonstrate consistent performance gains obtained by our Deep Fourier Shifting while reducing the computation burden. Additionally, ablation studies verify the robustness of the shift displacement with stable performance improvement.

## 1 Introduction

The spatial shift operator [3] is a technique that shifts channels from a pixel to its adjacent pixels. It is celebrated for its efficient facilitation of spatial information exchange. Due to its parameter-free nature and computational efficiency, this operator has found widespread application as a substitute for standard convolution units, particularly in high-level image classification tasks. A representative work, $S^2$-MLP [1], integrates the spatial shift operator into a channel-mixing MLP framework. This approach serves as an alternative to token-mixing MLPs, effectively mitigating intrinsic overfitting issues and significantly enhancing recognition accuracy. ShiftVit [2] explores the role of the self-attention mechanism in Vision Transformers (ViTs) for high-level tasks. It proposes a replacement using a modified spatial shift operation [4] that facilitates the exchange of a subset of channels

38th Conference on Neural Information Processing Systems (NeurIPS 2024).

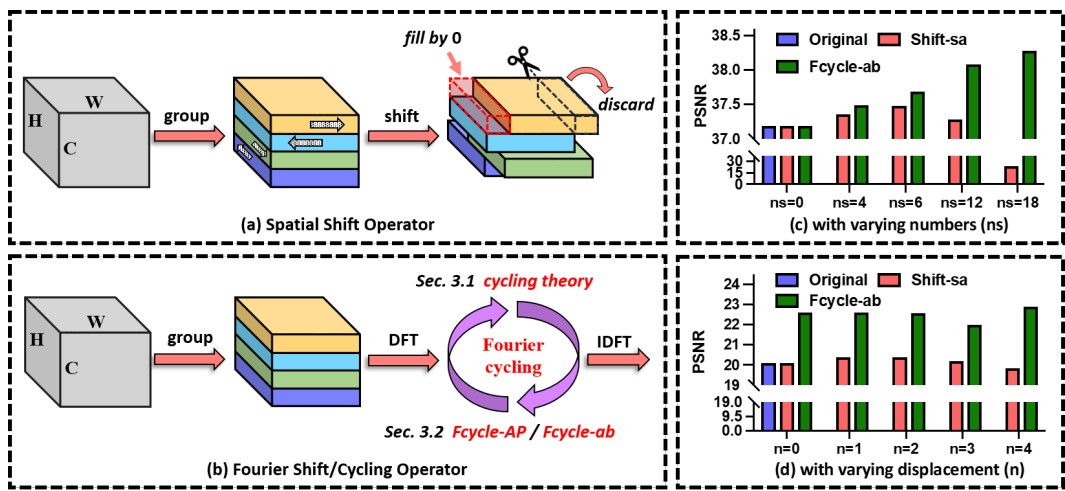

Figure 1: **Comparison between the spatial shift operator and the proposed deep Fourier shift operator.** (a) Traditional spatial shift operator involves a spatial shift mechanism that moves each channel of the input tensor in a distinct spatial direction, thus suffering from severe region-aware information loss and conflicting with the requirements of image restoration tasks. (b) Deep Fourier Shifting/Cycling operator is a more ingenious information-lossless operator, which is tailored for image restoration tasks. (c), (d) Deep Fourier shifting achieves a more stable performance gain than the spatial shifting mechanism with varying "ns" shift displacements and "n" basic units over image de-noising task where the cut-off is for compressing the vertical axis scale to better illustrate the contrast effect clearly.

between neighboring features. Despite these impressive achievements, the utility of spatial shift operators in low-level image restoration tasks remains unexplored.

The main difficulty in introducing the spatial shift mechanism to low-level vision tasks lies in its inherent conflict with the objectives of image restoration tasks, as depicted in Figure 1. In particular, this mechanism operates by moving each channel of its input tensor in distinct spatial directions, leading to a mixing of spatial information across channels. Notably, image restoration is fundamentally a standard regression problem where both features and channels play critical roles in determining the final output. This contrasts with the behavior observed in high-level vision tasks. A key limitation of the spatial shift operator is its inherent loss of region-aware information, where the regions subject to shifting are filled with zero values. Consequently, image restoration tasks, which are particularly sensitive to this kind of information loss, can experience a decline in performance due to spatial shifting.

In response to these challenges, we introduce a novel deep shifting operator specifically crafted for low-level image restoration tasks, named **Deep Fourier Shifting**. This operator revisits and enhances the fundamental principles of the traditional shift operator by extending its application into the Fourier domain. Here, the shift operation is reformulated as an *information-lossless* process executed through Fourier cycling. Deep Fourier Shifting is composed of two distinct variants, each featuring three key elements: a 2D discrete Fourier transform, Fourier cycling rules, and a 2D inverse Fourier transform. This operator is designed to be generic, and seamlessly integrable into existing image restoration architectures as a replacement for standard convolution units, offering the added advantage of reduced parameter usage. To assess the effectiveness of Deep Fourier Shifting, we undertake comprehensive experiments across a spectrum of low-level image restoration tasks. These include image denoising, low-light image enhancement, guided image super-resolution, and image de-blurring. Our experiments consistently reveal performance improvements achieved through the integration of Deep Fourier Shifting, while alleviating computational burden. Furthermore, we conduct ablation studies to evaluate the robustness of Deep Fourier Shifting against varying shift displacements. These studies demonstrate its ability to consistently enhance performance, underscoring its stability and effectiveness.

Our results suggest that Deep Fourier Shifting is a promising tool for image restoration, showing potential for varied real-world applications. We hope that Deep Fourier Shifting could contribute

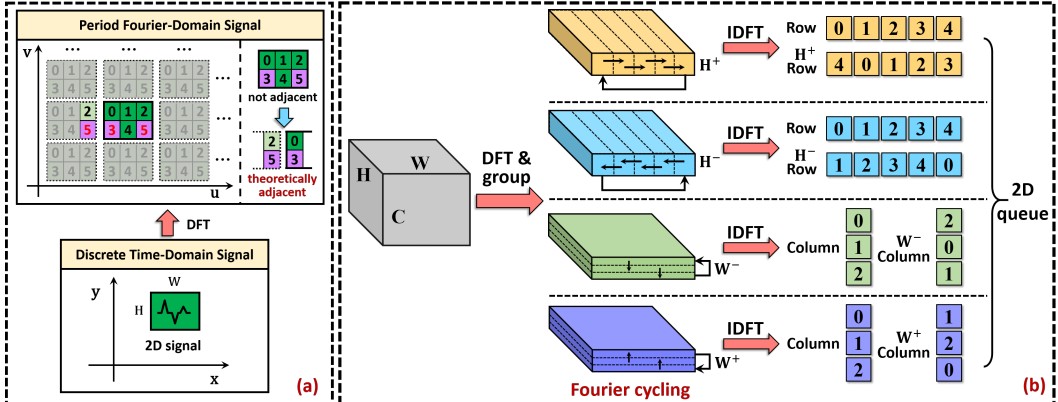

Figure 2: **(a) The information-lossless cycling mechanism.** The discrete Fourier transform of a signal exhibits period-extended and cycling properties. Specifically, in the Fourier domain, the two pixels in sequence beginning and end may not appear adjacent, but due to the period property, they are actually considered adjacent, as indicated by the upper right corner. This inherent period-extended and cycling behavior of the Fourier transform enables us to model the shifting mechanism in a manner that is information-lossless, making it well-suited for image restoration tasks. Consider the Fourier transform of a discrete time-domain signal, represented as $\left(\begin{smallmatrix} 0 & 1 & 2 \\ 3 & 4 & 5 \end{smallmatrix}\right)$. It may appear that the values 3 and 5 are not adjacent within the main period. However, owing to the property of period extension, the 3 from the previous period and the 5 from the current period are theoretically considered adjacent. It is reasonable to move the removed area from the end to the front, meeting the cycling mechanism. **(b) Our deep Fourier shifting operator.** Our operator borrows the principle of the spatial shifting mechanism and models the shifting mechanism in information-lossless Fourier cycling rules. The cycling is coded as 2D queue rolling.

to advancements in neural network designs for image restoration, particularly in improving spatial communication interactions.

## 2 Deep Fourier Shifting

Building on the principle of the shift operator, we extend its application to the Fourier domain. In this domain, the shift operator is conceptualized as an information-lossless operation, which we term Fourier cycling. To substantiate this approach, we present a theorem along with its corresponding proof. Additionally, we introduce two distinct variants of Deep Fourier shifting: i) the magnitude and phase variant, and ii) the real and imaginary variant, which are derived from the transformation rules we have identified within the Fourier domain.

**Definitions.** $f(x, y) \in \mathbb{R}^{H \times W \times C}$ is the spatial signal and $F(u, v) \in \mathbb{R}^{H \times W \times C}$ denotes its Fourier transform where $(x, y)$ and $(u, v)$ represent the space coordinates and Fourier spectrum, respectively.

**Theorem.** *The Fourier transform of a discrete signal is a period-extended and cycling:* $F(u, v) = F(u + nH, v) = F(u, v + mW) = F(u + nH, v + mW)$ *where* $u = 0, 1, 2, \ldots, H - 1$, $v = 0, 1, 2, \ldots, W - 1$ *and* $n, m \in \mathbb{N}$. $\mathbb{N}$ *is the set of positive integers starting from zero.*

### 2.1 Proof: The Fourier Transform of a Discrete Signal is Period-Extended and Cycling

We show the periodicity and cycling properties of the Fourier transform of a discrete signal, as illustrated in Figure 2(a). Note that the Fourier transform $F(u, v)$ of $f(x, y)$ is expressed as

$$F(u, v) = \frac{1}{HW} \sum_{x=0}^{H-1} \sum_{y=0}^{W-1} f(x, y) e^{-j2\pi\left(\frac{ux}{H} + \frac{vy}{W}\right)}. \tag{1}$$

Then, we show the periodicity of $F(u, v) \in \mathbb{R}^{H \times W}$ with H and W. It means $F(u, v) = F(u + nH, v) = F(u, v + mW) = F(u + nH, v + mW)$ where $u = 0, 1, 2, \ldots, H - 1$,

$v = 0, 1, 2, \ldots, W - 1$ and $n, m \in \mathbb{N}$ that records the set of non-negative integers. We take the $F(u, v) = F(u + n\mathrm{H}, v + m\mathrm{W})$ for example and recall Eq. (1) as

$$
\begin{aligned}
F(u &+ n\mathrm{H}, v + m\mathrm{W}) \\
&= \frac{1}{\mathrm{HW}} \sum_{x=0}^{\mathrm{H}-1} \sum_{y=0}^{\mathrm{W}-1} f(x, y).e^{-j2\pi(\frac{(u+n\mathrm{H})x}{\mathrm{H}} + \frac{(v+m\mathrm{W})y}{\mathrm{W}})} \\
&= \frac{1}{\mathrm{HW}} \sum_{x=0}^{\mathrm{H}-1} \sum_{y=0}^{\mathrm{W}-1} f(x, y)e^{-j2\pi(\frac{ux}{\mathrm{H}} + \frac{vy}{\mathrm{W}})} e^{-2j\pi m} e^{-2j\pi n} \\
&= F(u, v)e^{-2j\pi m} e^{-2j\pi n},
\end{aligned}
\tag{2}
$$

where for any integer $z$, it has $e^{-2j\pi z} = 1$.

Further, $e^{-2j\pi n} = 1$ and $e^{-2j\pi m} = 1$ for $n, m \in \mathbb{N}$. Therefore,

$$
F(u + n\mathrm{H}, v + m\mathrm{W}) = F(u, v)e^{-2j\pi m} e^{-2j\pi n} = F(u, v).
\tag{3}
$$

Similarly, we can prove the periodicity of $F(u, v)$ as well.

$$
\begin{aligned}
F(u, v) &= F(u + n\mathrm{H}, v + m\mathrm{W}) \\
&= F(u + n\mathrm{H}, v) = F(u, v + m\mathrm{W}).
\end{aligned}
\tag{4}
$$

Furthermore, deep Fourier transform can be expressed in Cartesian and polar coordinates by an equivalent form as

$$
F(u, v) = \mathrm{A}e^{j\mathrm{P}} = a + bj.
\tag{5}
$$

The period-extended and cycling property holds over the amplitude-phase and real-imaginary format.

## 2.2 Architectural Design

Recall the **Theorem-1**, we propose two deep Fourier shifting variants: amplitude-phase variant and real-imaginary variant.

**Amplitude-phase shifting variant.** This variant is illustrated in Figure 2(b). The pseudo-code is shown in Figure 3 (left). Given an image $\mathrm{X} \in \mathbb{R}^{\mathrm{H} \times \mathrm{W} \times \mathrm{C}}$, we first adopt the Fourier transform $\mathtt{FFT(X)}$ to obtain its amplitude component $\mathtt{A}$ and phase component $\mathtt{P}$. We then evenly split the generated $\mathtt{A}$ and $\mathtt{P}$ in 4 folds $\mathtt{A\_g}$ and $\mathtt{P\_g}$ by the channel dimension, perform the Fourier cycling over $\mathtt{A\_g}$ and $\mathtt{P\_g}$ in both the H and W dimensions:

$$
\begin{aligned}
\mathtt{A\_g[0]} &= \mathtt{torch.roll(A\_g[0], dim{=}1, s)} \\
\mathtt{A\_g[1]} &= \mathtt{torch.roll(A\_g[1], dim{=}1, \text{-}s)} \\
\mathtt{A\_g[2]} &= \mathtt{torch.roll(A\_g[0], dim{=}2, s)} \\
\mathtt{A\_g[3]} &= \mathtt{torch.roll(A\_g[0], dim{=}2, \text{-}s)},
\end{aligned}
\tag{6}
$$

where $\mathtt{torch.roll(.)}$ accounts for the cycling function by $\mathtt{dim}$ parameter for shifting dimension and $\mathtt{s}$ for shifting displacement. The transformed $\mathtt{A\_g}$ and $\mathtt{P\_g}$ are then fed into two independent convolution modules with $1 \times 1$ kernel and followed by the inverse Fourier transform $\mathtt{iFFT(.)}$ to project the shifting ones back to spatial domain.

**Real-imaginary shifting variant.** The pseudo-code for the real-imaginary shifting variant is presented on the right side of Figure 3. In this variant, we perform Fourier cycling separately on the real component $a$ and the imaginary component $b$, while keeping the remaining processing steps the same as the amplitude-phase shifting variant.

Concerning two variants, the first one entails trigonometric function calculations, wherein minor numerical alterations could potentially result in computational instability in engineering applications. However, it offers more accurate physical interpretations over amplitude-phase operation from signal processing perspective.

```python
def DFS_AP(X):                               def DFS_ab(X):
 # X: input with shape [N, C, H, W]           # X: input with shape [N, C, H, W]
 # A and P are the amplitude and phase        # a and b are the real and imaginary part
    A.e{jP} = FFT(X)                             a+bj = FFT(X)

    # Fourier shifting transform rules            # Fourier shifting transform rules
    A_g = torch.spilt(A, 4, dim=1)               a_g = torch.spilt(a, 4, dim=1)
    P_g = torch.spilt(P, 4, dim=1)               b_g = torch.spilt(b, 4, dim=1)
    A_fc = Fourier-cycling(A_g)                  a_fc = Fourier-cycling(a_g)
    P_fc = Fourier-cycling(P_g)                  b_fc = Fourier-cycling(b_g)
    A_fc = Convs_1x1(A_fc)                       A_fc = Convs_1x1(a_fc)
    P_fc = Convs_1x1(P_fc)                       P_fc = Convs_1x1(b_fc)

    # Inverse Fourier transform                   # Inverse Fourier transform
    Y = iFFT(A_fc, P_fc)                          Y = iFFT(a_fc, b_fc)

    Return Y #[N, C, H, W]                       Return Y #[N, C, H, W]
```

Figure 3: **Pseudo-code of the two variants of the proposed deep Fourier shifting.** The left is the *amplitude-phase variant* while the right is the *real-imaginary variant*.

Table 1: Quantitative comparisons on low-light image enhancement. The arrow $\rightarrow$ denotes the generalization setting by training on the data before the arrow and testing directly on the data after the arrow.

| Model | Config | LOL $\rightarrow$ | | $\rightarrow$ Huawei | | Huawei $\rightarrow$ | | $\rightarrow$ LOL | | #Paras |
| | | PSNR | SSIM | PSNR | SSIM | PSNR | SSIM | PSNR | SSIM | |
|---|---|---|---|---|---|---|---|---|---|---|
| DRBN | Original | 19.7931 | 0.8361 | 17.7929 | 0.6247 | 20.1549 | 0.6851 | 18.0856 | 0.7543 | 0.55M |
| | Shift-sa | 19.7072 | 0.8343 | 17.6221 | 0.6071 | 20.2165 | 0.6873 | 17.9112 | 0.7532 | 0.41M |
| | Fcycle-AP | **22.4274** | **0.8448** | **19.3252** | **0.6472** | 20.5855 | 0.6872 | 18.8666 | 0.7587 | 0.41M |
| | Fcycle-ab | 22.2054 | 0.8429 | 19.3125 | 0.6431 | **20.6651** | **0.6876** | **19.1535** | **0.7681** | 0.41M |
| SID | Original | 20.1062 | 0.7895 | 16.5874 | 0.5925 | 20.1742 | 0.6659 | 18.5468 | 0.7441 | 7.76M |
| | Shift-sa | 20.0148 | 0.7911 | 16.8214 | 0.5911 | 20.1517 | 0.6651 | 18.4998 | 0.7434 | 7.53M |
| | Fcycle-AP | **22.8565** | **0.8019** | 19.1707 | 0.6238 | 20.9068 | **0.6708** | **18.8161** | **0.7494** | 7.53M |
| | Fcycle-ab | 22.6313 | 0.7995 | **19.2471** | **0.6242** | **20.9271** | 0.6691 | 18.5741 | 0.7443 | 7.53M |

Table 2: Comparisons on image denoising.

| Dataset | Metric | DNCNN | | | |
| | | Original | Shift-sa | Fcycle-AP | Fcycle-ab |
|---|---|---|---|---|---|
| SIDD | PSNR | 37.1992 | 37.2247 | 37.6891 | **38.1837** |
| | SSIM | 0.8954 | 0.8980 | **0.9013** | 0.9066 |
| DND | PSNR | 38.33 | 38.42 | **38.69** | 38.65 |
| | SSIM | 0.8974 | 0.8985 | 0.942 | **0.949** |
| | #Paras | 1.51M | 1.43M | 1.43M | 1.43M |

Table 3: Comparisons on image deblurring.

| Dataset | Metric | DeepDeblur | | | |
| | | Original | Shift-sa | Fcycle-AP | Fcycle-ab |
|---|---|---|---|---|---|
| GoPro | PSNR | 28.9423 | 28.9037 | **29.2123** | 29.1939 |
| | SSIM | 0.8716 | 0.8712 | **0.8777** | 0.8765 |
| HIDE | PSNR | 26.9770 | 26.9991 | **27.2860** | 27.2547 |
| | SSIM | 0.8468 | 0.8476 | **0.8541** | 0.8536 |
| | #Paras | 11.72M | 10.61M | 10.61M | 10.61M |

## 3 Experiments

### 3.1 Experimental Settings

**Image enhancement.** We evaluate our Fourier shifting operator on two popular image enhancement benchmarks: LOL [5] and Huawei [6]. The LOL dataset consists of 500 low-/normal-light image pairs. Following the original setting, we use 485 pairs for training and 15 pairs for testing. The Huawei dataset contains 2480 paired images, with 2200 pairs for training and 280 pairs for testing. We compare with the representative approaches SID [7] and DRBN [8].

**Image deblurring and denoising.** For the image deblurring task, we employ DeepDeblur [9] in our experiments. We use the GoPro dataset [9] for training. To demonstrate the generalizability of our operator, we also apply the model trained on the GoPro dataset directly to the test images of the HIDE dataset [10]. For image denoising, we use the SIDD dataset [11] as the training benchmark.

Table 4: Quantitative comparisons on guided image super-resolution.

| Model | Config | WorldView-II | | | | GaoFen2 | | | | WorldView-III | | | |
|---|---|---|---|---|---|---|---|---|---|---|---|---|---|
| | | PSNR | SSIM | SAM | ERGAS | PSNR | SSIM | SAM | EGAS | PSNR | SSIM | SAM | EGAS |
| PANNET | Original | 40.8172 | 0.9630 | 0.0257 | 1.0555 | 42.1699 | 0.9569 | 0.0192 | 0.9565 | 29.68 | 0.907 | 0.085 | 3.426 |
| | Shift-sa | 40.8791 | 0.9631 | 0.0255 | 1.0511 | 42.2107 | 0.9577 | 0.0185 | 0.9549 | 29.32 | 0.897 | 0.103 | 3.734 |
| | Fcycle-AP | 41.2633 | 0.9650 | 0.0242 | 1.0080 | **42.6361** | **0.9680** | **0.0173** | **0.9005** | 30.46 | 0.915 | 0.078 | 3.253 |
| | Fcycle-ab | **41.3184** | **0.9671** | **0.0238** | **1.0032** | 42.5594 | 0.9648 | 0.0185 | 0.9118 | 30.55 | 0.918 | 0.077 | 3.187 |
| MutNet | Original | 41.4967 | 0.9692 | 0.0232 | 0.9781 | 47.1069 | 0.9883 | 0.0106 | 0.5626 | 30.59 | 0.924 | 0.0741 | 3.0798 |
| | Shift-sa | 41.5227 | 0.9695 | 0.0230 | **0.9766** | 47.1371 | 0.9886 | 0.0098 | 0.5617 | 29.87 | 0.912 | 0.082 | 3.325 |
| | Fcycle-AP | **41.6686** | **0.9712** | 0.0228 | 0.9598 | **47.4225** | **0.9890** | 0.0101 | **0.5453** | 30.71 | 0.926 | 0.0737 | 3.0767 |
| | Fcycle-ab | 41.6579 | 0.9701 | 0.0228 | 0.9611 | 47.3489 | 0.9887 | 0.0104 | 0.5479 | **30.74** | **0.926** | **0.0737** | **3.0764** |

Performance evaluation is conducted on the remaining validation samples from the SIDD dataset and the DND benchmark dataset [12]. The selected baseline for comparison is DnCNN [1] [13].

**Guided image super-resolution.** We adopt the pan-sharpening task as a representative task of guided image super-resolution. The WorldView II and GaoFen2 datasets [14] are used in our experiments. The baselines are state-of-the-art INNformer [15] and SFINet [14].

In evaluating the performance of different approaches, we use image quality assessment metrics such as the relative dimensionless global error in synthesis (ERGAS) [16], the peak signal-to-noise ratio (PSNR), structural similarity index (SSIM), the spectral angle mapper (SAM) [17].

## 3.2 Implementation Details

Based on the above competitive baselines, we perform the comparison over the following configurations by replacing the standard convolution with the spatial or Fourier shifting operator:

1) **Original**: the baseline without any changes;
2) **Fcycle-AP**: replacing the original model's standard convolution operator with the amplitude-phase variant of Deep Fourier shifting;
3) **Fcycle-ab**: replacing the original model's standard convolution operator with the real-imaginary variant of Deep Fourier shifting;
4) **Shift-sa**: replacing the variants of Deep Fourier shifting in the settings of $2)/3)$ with the spatial shifting operator;
5) **Shift-ns**: replacing "ns" standard convolution operator with the spatial shifting or Fourier shifting-ab/AP operator;
6) **Shift-n**: replacing the spatial shifting or Fourier shifting-ab/AP operator in the settings of $5)$ with varying shifting displacement "n".

## 3.3 Comparison and Analysis

**Quantitative comparison.** We evaluate model performance across various configurations as detailed in the implementation section (Section 3.2). The quantitative results of this analysis are systematically presented in Tables 1, 2, 3, and 4. In these tables, the best and second-best results are highlighted in bold and underlined, respectively. Values that fall below the established baseline are marked with a gray background for clear distinction. We observe a uniform trend of performance enhancement across all the tasks and datasets tested when our two novel deep Fourier shifting variants are integrated. This improvement is particularly noteworthy as it is achieved with reduced computational costs, suggesting the efficiency and effectiveness of our approach. For instance, in the low-light image enhancement baseline SID (Table 1), our "Fcycle-AP" and "Fcycle-ab" models surpass the "Original" model by achieving higher PSNR values of 2.75dB/2.53dB and 2.6dB/2.7dB on the LOL and Huawei datasets, respectively. In contrast, the integration of the spatial shifting operator into the SID baseline results in a slight reduction in PSNR by 0.1dB/0.02dB on the LOL and Huawei datasets, as evident in the 3rd and 7th rows. These findings validate the superiority of our proposed Fourier shifting approach, demonstrating its suitability and effectiveness in addressing the unique challenges of low-level image restoration tasks. Additionally, the integration of our proposed methods has shown to enhance the training performance, in Figure 4.

---

[1] We followed the official code to reproduce the results `https://github.com/cszn/KAIR`.

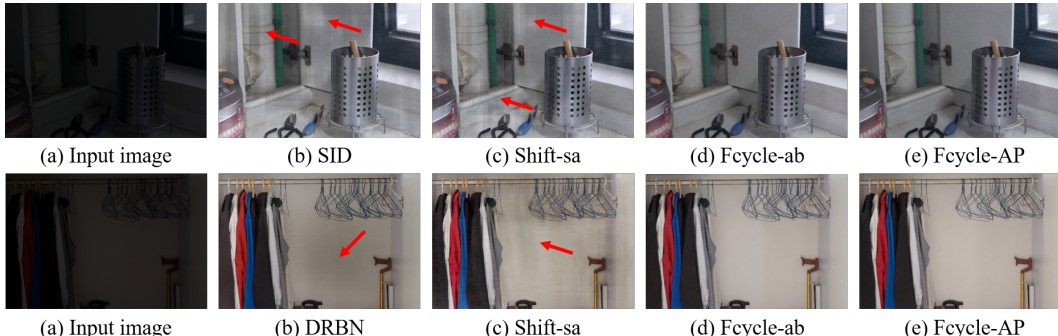

(a) Input image | (b) SID | (c) Shift-sa | (d) Fcycle-ab | (e) Fcycle-AP

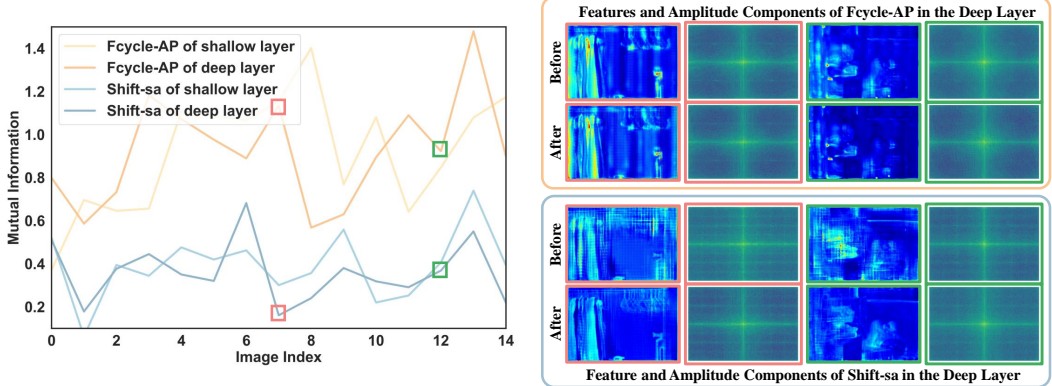

(a) Input image | (b) DRBN | (c) Shift-sa | (d) Fcycle-ab | (e) Fcycle-AP

Figure 5: Visual comparison over image enhancement.

Figure 6: The effectiveness of information preservation. **Left:** we compare mutual information levels before and after employing Fcycle-ab and Shift-sa operators on the LOL test set, respectively. Our operator exhibits significantly higher mutual information than Shift-sa, showcasing its efficacy in information preservation. **Right:** we visualize feature maps and their amplitude components before and after operations. This demonstrates that our Fcycle-AP promotes frequency information i̶  l domain.

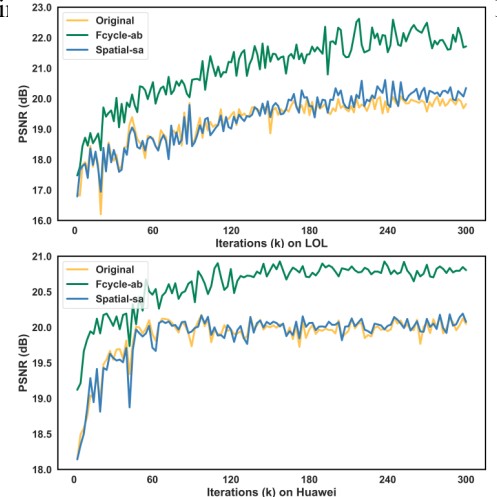

Figure 4: The proposed operators improve the training performance. It shows the training PSNR on the image enhancement task on the LOL and Huawei datasets in the top and bottom.

**Qualitative comparison.** Due to space constraints, we present only a subset of the visual results in Figure 5, which effectively illustrate the efficacy of our proposed deep Fourier shifting operator. Additional visual results are available in the supplementary materials. As demonstrated in these figures, the integration of the deep Fourier shifting operators (Fcycle-ab and Fcycle-AP) with the original baseline models yields results that are visually more appealing compared to the baselines and the spatial shift operator (Shift-sa). Models enhanced with our operators exhibit a superior ability to restore fine texture details and mitigate degradation effects. In contrast, models using the spatial shift operator tend to produce significant artifacts.

**Information preservation.** To offer a deeper understanding of the efficacy of our frequency cycling mechanism, we begin by comparing the mutual information of features before and after the application of our proposed Fcycle-AP operator on the LOL test set. The statistical analysis, illustrated in Figure 6 (left), demonstrates a signifi-

| Model | Config | Fcycle-ab | | Fcycle-AP | | Shift-sa | | #Paras |
|---|---|---|---|---|---|---|---|---|
| | | PSNR | SSIM | PSNR | SSIM | PSNR | SSIM | |
| Original | #0 | 37.1992 | 0.8954 | 37.1992 | 0.8954 | 37.1992 | 0.8954 | 1.51M |
| Shift-ns | #2 | 37.2309 | 0.8958 | 37.3781 | 0.8983 | 37.2247 | 0.8980 | 1.43M |
| | #4 | 37.3981 | 0.8974 | 37.4832 | 0.8981 | 37.4256 | 0.8994 | 1.35M |
| | #6 | 37.6964 | 0.9011 | **37.5891** | **0.9013** | **37.5091** | **0.8993** | 1.28M |
| | #12 | 38.0837 | 0.9064 | 37.4676 | 0.9003 | 37.2996 | 0.8983 | 1.06M |
| | #18 | **38.2810** | **0.9081** | 37.5251 | 0.9044 | 23.6867 | 0.3340 | 0.84M |
| Shift-n | #1 | 37.2309 | 0.8958 | 37.3781 | 0.8983 | 37.2247 | 0.8980 | 1.43M |
| | #2 | 37.2663 | 0.8961 | 37.3551 | 0.8978 | 37.3379 | 0.8995 | 1.43M |
| | #3 | **37.4277** | **0.8985** | 37.3139 | 0.8974 | **37.4009** | **0.8998** | 1.43M |
| | #4 | 37.3781 | 0.8977 | **37.4754** | **0.9001** | 37.1247 | 0.8909 | 1.43M |

Table 5: Ablation studies of image denoising network, DNCNN.

cantly higher level of mutual information as outlined in [18] for our method when compared to the conventional shifting operation in the spatial domain. This notable increase in mutual information underscores the strength of our Fourier cycling mechanism in preserving information and minimizing information loss.

Moreover, an analysis of the features and their corresponding amplitude components, as depicted in Figure 6 (right), reveals distinct grid effects in the feature maps generated by the Shift-sa operator. In contrast, our method substantially mitigates these grid effects. This is achieved by facilitating interactions among frequency information and improving spatial communication. Together, these observations compellingly show the efficiency of our proposed Fourier cycling mechanism in both preserving information and enhancing frequency interaction.

## 3.4 Ablation Studies

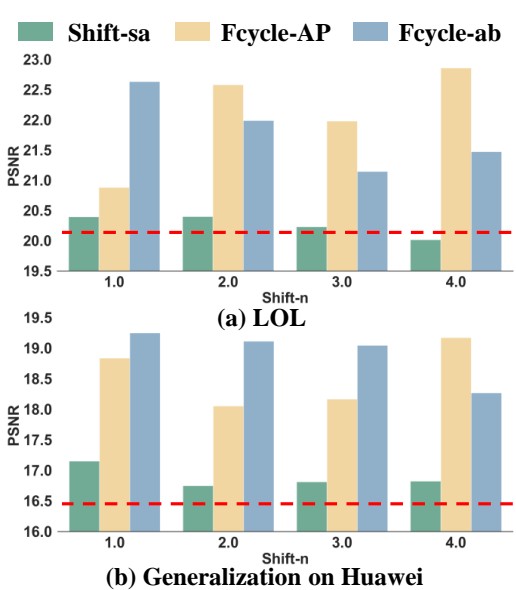

**(a) LOL**

**(b) Generalization on Huawei**

Figure 7: The effect of shifting displacement shift-n on SID.

**With varying "n".** To assess the robustness of our method, we conduct a comparative analysis with the image enhancement baseline SID and the image denoising baseline DNCNN. This comparison involved varying the shifting displacement "n". The corresponding quantitative results are presented in Figure 7 and Table 5. The results clearly show that integrating our proposed Fourier shifting solutions into these baseline models consistently yields better performance than the original baselines alone. Conversely, when spatial shifting is employed within these baselines, there is a marked decline in performance compared to their original versions, particularly at larger shifting displacements (e.g., "n=4"). These findings not only validate the efficacy of our approach but also highlight its distinct robustness over traditional spatial shifting.

**With varying "ns".** We examine the robustness of our deep Fourier shifting variants by comparing them with the image de-noising baseline DNCNN. For this comparison, we substitute the variable "ns" standard convolution units in the baseline with both the spatial shifting operator or our proposed deep Fourier shifting variants, as detailed in Section 3.2. The results of this evaluation are provided in Table 5. We highlighted the best and second-best results in bold and underline, respectively. A visual depiction of these results is provided in Figure 1 (c). A key observation from these results is that as "ns" increases, models incorporating our proposed operators not only

exhibit consistent performance improvements but also achieve these enhancements with fewer model parameters. In contrast, models using the spatial shifting operator display a notable decrease in performance, particularly when "ns" is increased to 18. This trend demonstrates the robustness and efficiency of our proposed Fourier shifting solutions. To further elucidate this point, values falling below the baseline are marked with a gray background, highlighting their relevance and impact.

## 4 Related Work

**Spatial-shifting operator.** The spatial-shift operator, initially proposed by Wu et al.[3], enhances efficient spatial information communication. It has gained traction in high-level image classification tasks [1], where they innovated upon MLPMixer [19]. They replaced the spatial-wise token mixer with a spatial-shift operation, enabling inter-patch communication and effectively addressing overfitting challenges associated with spatial-specific token-mixing MLPs. Similarly, Wang et al. [2] explored the role of attention mechanisms in Vision Transformers (ViTs). They introduced a partial shift operation, exchanging a subset of channels among adjacent features. This simple yet effective operation led to the development of ShiftViT, a backbone network that replaces traditional attention layers in ViTs with shift operations. These approaches illustrate the adaptability and versatility of spatial-shift operators in enhancing deep learning models for image analysis.

**Deep Fourier transform over image restoration.** There has been a growing interest in integrating the Fourier transform to refine deep learning-based image restoration models. A notable example is the work of Zhou et al. [20], which re-examines the interplay between spatial and Fourier domains. This study uncovers transformation rules applicable to various resolution features within the Fourier domain, leading to the development of a theoretically sound Deep Fourier Up-Sampling method applicable across multiple restoration tasks. Similarly, the research in Zhou et al. [14] investigates the degradation dynamics of guided image super-resolution. The authors propose a novel spatial-frequency dual-domain integration network tailored to this specific task. The idea leads to a deep Fourier-based exposure correction network designed to tackle exposure-related issues. These pioneering studies harness the synergistic potential of deep learning and the intrinsic attributes of the Fourier transform.

## 5 Limitations

Our study acknowledges that there is room for more comprehensive experiments and exploration of representative baselines on broader computer vision tasks. It is important to note that this work represents the initial endeavor to delve into the utilization of shifting mechanisms and the development of tailored deep Fourier shifting operators specifically for low-level image restoration tasks. Moreover, our focus extends beyond designing a plug-and-play module for integration into existing networks to achieve performance improvements. We aim to provide a powerful and efficient spatial communication interaction choice by offering an alternative to the basic convolution operator pool when constructing new models from scratch.

## Broader Impact

Our research has the potential to facilitate efficient image restoration on edge devices and resource-limited platforms, democratizing access to high-quality capabilities. This benefits critical domains like mobile photography, remote sensing, and medical imaging. Additionally, reduced computational requirements contribute to energy efficiency, extending battery life and reducing environmental impact. We emphasize responsible development, evaluation, and deployment to ensure equitable and ethical use of our methods.

## 6 Conclusion

We have presented Deep Fourier Shifting, a shifting operator grounded in solid theoretical principles, specifically tailored for image restoration tasks. This operator, characterized by its information-lossless nature, leverages the Fourier cycling approach to shift operations. A key feature of our Deep Fourier Shifting is its versatility; it can be seamlessly integrated as a replacement for standard convolution units in existing image restoration networks, offering the added benefit of reduced parameter usage. Our experimental evaluations have consistently demonstrated the enhancements in performance attributable to the incorporation of our Deep Fourier Shifting.

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
