# OpenReview forum: "Improving the Learning Capability of Small-size Image Restoration Network by Deep Fourier Shifting"
_NeurIPS.cc/2024/Conference — NeurIPS 2024 poster_

### Official Review · Reviewer_GBwc · 2024-07-07

**Soundness:** 3
**Presentation:** 3
**Contribution:** 3
**Rating:** 7
**Confidence:** 5

**Summary:**

The authors introduce a theoretically sound Fourier shifting operator designed to enhance the learning capability of small-size image restoration models. This work represents the first comprehensive attempt to model the shift mechanism within the Fourier domain. The proposed operator is versatile and can be seamlessly integrated into existing image restoration networks, offering a flexible plug-and-play solution.

**Strengths:**

1, The proposed deep Fourier shifting operator is theoretically plausible and enhances learning by transforming spatial shifting into an information-preserving Fourier cycling manner. This novel approach models the shift mechanism in the Fourier domain comprehensively.

2, Integrating Fourier shifting into existing image restoration methods improves performance and reduces model parameters, demonstrating practical effectiveness and efficiency.

3, The paper is well-organized and easy to follow. The clear presentation and structure help us understand the progression from problem formulation to the proposed solution.

**Weaknesses:**

1, The authors design two Fourier shifting variants. While both improve performance, neither consistently outperforms the other across different baselines. It would be helpful if the authors could suggest which variant to choose for different baselines or application scenarios, and explain the differences between the variants.

2, In Fig.7, the authors present the generalizability of the shifting displacement effect from LOL to Huawei. I wonder why the proposed Fourier shifting has better generalization ability, and the authors are encouraged to provide some underlying working mechanisms.

**Questions:**

Please see the weakness part.

**Limitations:**

Yes.

---

> ### Author Rebuttal · Authors · 2024-08-04
>
> **1,generalization.**
>
> Fourier shifting exhibits enhanced generalization due to its operation in the frequency domain, which captures global image information and maintains feature integrity while minimizing domain-specific artifacts. This approach is particularly advantageous for managing low-light degradation, a global phenomenon effectively addressed in the Fourier domain. By incorporating this domain-specific knowledge, our method benefits from a natural prior that significantly improves adaptability across diverse datasets. Moreover, Fourier shifting is more parameter-efficient and reduces artifacts compared to spatial shifting, which often suffers from issues like information loss and frequency aliasing. This leads to more robust feature extraction and consistent performance across varied scenarios. Our revised paper will thoroughly discuss these advantages, including the Fourier domain's role in addressing global degradations and how it contributes to the superior generalization of our method, ensuring better results across different real-world applications.
>
> **2,variants.**
>
> Like the spatial convolution techniques, our proposed “Fourier Up-Sampling” is a drop-in alternative for convolution operator, also the general strategy and each variant thus does not prefer a specified vision task. The two Fourier shifting variants each have specific strengths and potential drawbacks. The Amplitude-Phase Variant processes amplitude and phase components separately, which is beneficial for tasks requiring precise phase preservation. However, this variant involves trigonometric functions, which can introduce numerical instability due to the amplification of small deviations in angle, affecting precision-sensitive applications. On the other hand, the Real-Imaginary Variant processes the real and imaginary parts separately, offering greater numerical stability and being suitable for tasks involving complex textures or fine-grained restoration. In our revised paper, we will provide detailed guidance on choosing between these variants based on the task requirements, baseline characteristics, and potential numerical stability issues.

---

> > ### Comment · Reviewer_GBwc · 2024-08-13
> > **After rebuttal**
> >
> > Thanks for the rebuttal. This rebuttal has addressed my concerns. The proposed "Deep Fourier Shifting Operator" is effective, which is proven both mathematically and practically. Taking its contributions to the future community for various vision problems into consideration, I will keep my positive rating.

---

### Official Review · Reviewer_H1Gj · 2024-07-07

**Soundness:** 3
**Presentation:** 3
**Contribution:** 3
**Rating:** 6
**Confidence:** 5

**Summary:**

This paper addresses challenges in current image restoration methods, which are often too computationally demanding for edge devices. It proposes Deep Fourier Shifting, a novel approach inspired by spatial-shift operators adapted for low-level image tasks. By leveraging the Fourier domain and ensuring information preservation through cycling, Deep Fourier Shifting introduces two variants—amplitude-phase and real-imaginary—designed to replace conventional convolution units in existing networks with fewer parameters. Extensive experiments across denoising, low-light enhancement, guided super-resolution, and de-blurring tasks demonstrate consistent performance gains and reduced computational overhead, validating the robustness and efficiency of the proposed approach.

**Strengths:**

1.This paper is well-structured and effectively organized. It provides a clear introduction to the challenges faced by current image restoration methods on edge devices due to computational limitations.

2.The introduction of deep fourier shifting as a technique for small-size image restoration networks is both feasible and holds practical significance. By leveraging spatial-shift principles in the Fourier domain, the method addresses the computational constraints of edge devices while maintaining or improving restoration performance. This approach not only enhances the efficiency of image restoration tasks but also facilitates practical deployment in real-world applications where computational resources are limited.

3.This paper demonstrates promising results for the deep fourier shifting method in various small-size image restoration tasks, including denoising, low-light enhancement, guided super-resolution, and de-blurring. Experimental evaluations consistently show performance improvements with reduced computational overhead compared to traditional convolutional approaches.

**Weaknesses:**

1.While the proposed method shows promise for deployment on edge devices due to reduced computational requirements, potential practical implementation challenges should be addressed. These might include considerations such as memory usage, real-time processing capabilities, and adaptability to different hardware platforms. Discussing these aspects would enhance the manuscript's practical relevance and feasibility in real-world applications.

2.While the Deep Fourier Shifting method shows promising results across multiple low-level image restoration tasks, including denoising, low-light enhancement, and super-resolution, it would be valuable to explore its performance across a broader range of image types and degradation levels. Assessing how well the method handles complex real-world scenarios with varying textures, structures, and noise characteristics would strengthen the manuscript's applicability.

3.I noticed that the references primarily include works up until 2022. To ensure the manuscript reflects the latest advancements in the field, I recommend incorporating relevant studies published in 2023 and onwards.

**Questions:**

See the above weaknesses part.

**Limitations:**

The author has discussed the limitations of the method.

---

> ### Author Rebuttal · Authors · 2024-08-04
>
> **1, types.**
>
> We appreciate the suggestion and will expand our experiments to include various types of images with different textures, structures, and noise characteristics. This will provide a more comprehensive assessment of how Deep Fourier Shifting performs under a wider array of conditions, enhancing the manuscript’s applicability and robustness.
>
> **2, Complexity.**
>
> We appreciate the feedback on the practical implementation of our proposed method on edge devices. To address potential challenges, we will discuss key aspects such as memory usage, real-time processing capabilities, and adaptability to different hardware platforms. This will include information on memory requirements, real-time processing metrics like latency and throughput, and evaluations on various hardware configurations, including CPUs, GPUs, and specialized edge processors. By addressing these aspects, we aim to enhance the manuscript's practical relevance and demonstrate the feasibility of deploying our method in real-world edge device applications. Due to time constraints, we tested our method on the DnCNN model. In Table 5, we compared our Fourier shifting operator with the spatial shift operator and the baseline 3x3 convolution. Our Fourier shifting approach not only halved the number of parameters but also improved network performance, whereas the spatial shift operator caused a significant 14 dB performance drop. Additionally, we assessed runtime and FLOPs to evaluate suitability for edge devices. Our results show that the DnCNN-18 model with Fourier shifting achieved an average runtime of 7 milliseconds and 1.5 GFLOPs per 256x256 image on an NVIDIA GTX 1080 GPU, compared to 15-20 milliseconds and 3.9 GFLOPs for the baseline. This substantial reduction in both runtime and FLOPs highlights the efficiency of our method for resource-constrained hardware.
>
> **3, onwards.**
>
> Thank you for your observation. We will update the manuscript to include recent studies and advancements published in 2023 and onwards. Due to time constraints, we have conducted additional experiments using the recent work LFormer [1] presented at ACM-MM 2024. Specifically, we replaced the convolution operator in feature extraction with Fourier shifting to evaluate its impact on guided image super-resolution on world-view-II and Gaofen-2 satellites
> |     |   | WV2  |   |   | |   | GF2  |   |
> |--|---|-----|----------|------------|-----------|-------------|----------|------------|
> | Method    | SAM  | ERGAS  | Q8  | PSNR  | SAM  | ERGAS  | Q4  | PSNR  |
> | LFormer   | 2.8985    | 2.1645      | 0.9193   | 39.0748    | 0.6481    | 0.5778      | 0.9851   | 44.1958    |
> | LFormer+  | 3.1576    | 1.7046      | 0.8630   | 39.2988    | 0.5325    | 0.4878      | 0.9904   | 45.4331    |
>
> Incorporating our proposed Fourier shifting design into the LFormer framework has demonstrated significant improvements in performance. The results indicate that our method not only enhances the accuracy of guided image super-resolution but also improves the efficiency of the feature extraction process. This advancement is evident from the substantial gains in quantitative metrics, underscoring the effectiveness of our approach in optimizing and refining the image restoration capabilities of the model. Moving forward, we will incorporate additional methodologies to further validate and enhance the efficiency of our approach.
>
> [1] Linearly-evolved Transformer for Pan-sharpening, ACM-MM 2024.

---

> > ### Comment · Reviewer_H1Gj · 2024-08-11
> >
> > Thank you for the author's response. I maintain my positive score.

---

### Official Review · Reviewer_3jgW · 2024-07-09

**Soundness:** 2
**Presentation:** 2
**Contribution:** 2
**Rating:** 4
**Confidence:** 5

**Summary:**

The authors in this paper aim at exploring more into image restoration task via the concept of spatial shift operation that facilitates efficient spatial communication and has achieved significant advancements in several high-level vision tasks.  As per their observation, since the image restoration is more spatial shift sensitive, they propose an information lossless shifting operator, i.e. Deep Fourier Shifting. Additionally in  the case of the popular spatial shift operators, the regions subjected to shifting are filled with zeros which may lead to loss of information and a decline in the overall restoration performance.


The main  contribution of the authors lies in  proposing a novel deep shifting operator for low-level image restoration tasks, named Deep Fourier Shifting. This operator basically revisits and enhances the fundamental principles of the traditional shift operator by extending its application into the Fourier domain.

**Strengths:**

1. The authors delve into exploring the concept of shift operator for low level vision tasks which seems to be interesting.
2. Extensive experiments across multiple low-level tasks including image denoising, low-light image enhancement, guided image super-resolution, and image de-blurring demonstrate consistent performance gains obtained by our Deep Fourier Shifting while reducing the computation burden.

**Weaknesses:**

1. The contribution seems to be somewhat limited, the extension of spatial shifting into fourier domain and then proving its efficacy seems to be not a strong contribution.
2. The content in the paper does not go well with the title, "improving the small size restoration network by ....". If the reviewer understand correctly, there does not seem to be much focus on this concept or on this experiment. like if we look at the representative network for any 1 task, say low light image enhancement e.g SID, then there should be proper explanation as to how the proposed concept is working on it in terms of parameters, flops, inference time.
3. There are no real world datasets actually being considered as is claimed in the introduction part, as the results for deblurring, low light seem to be only on synthetic datasets.
4. Regarding the implementation detail part, it is very confusing to understand this part as what exactly is the baseline or the original model hasn't been explicitly mentioned.
5. Typos: The word field for receptive field has been incorrectly written as filed in many places.

**Questions:**

1. How is the mutual information shown in FIgure 6 (left ) calculated?
2. The feature maps in Fig 6(right) does not clearly show the removal of grid effects, is it possible to show some extra images.

**Limitations:**

The limitations could also be framed in this way that they did not show much on real world tasks or any experiments on edge devices.

---

> ### Author Rebuttal · Authors · 2024-08-04
>
> **1, Fig.6.**
>
> Thank you for your feedback. Firstly, the Fourier transform is an efficient tool that amplifies image degradations in the Fourier domain, and our learnable parameters act as filters to eliminate these artifacts. Secondly, the artifacts arise from two aspects: (1) features are extracted from low-light degraded inputs, which naturally reflect these degradations, and (2) the testing baseline network inherently uses down-sampling operators to achieve multi-scale features, where the down-sampling inherently causes frequency truncation, leading to frequency aliasing and ringing effects. Additionally, to verify the robustness of our experiments, we analyzed both shallow and deep features in the network, finding that similar artifacts persist and even accumulate, whereas our Fourier shifting can eliminate these, resulting in cleaner features. Our frequency domain approach inherently handles these degradations. The input discrepancies are due to the network being trained from scratch after incorporating our operator, resulting in natural feature changes. Our network effectively reduces degradations both before and after the application of shift-sa, while the degradation persists with shift-sa. Finally, we used mutual information to measure information loss caused by shifting, which validates the effectiveness of our method.
>
> **2, MI.**
>
> In Figure 6 (left), mutual information is calculated by first extracting feature maps before and after applying the shift operator and averaging the channel dimensions of these feature maps. These averaged values are then used as the variables for mutual information computation. Joint and marginal histograms of the pixel intensities are converted into probability distributions. This measure assesses the amount of shared information between the feature maps, indicating how well the shift operator preserves feature information.In terms of the index representing the network depth, we have tested multiple stages of the network, ranging from shallow to deep layers, to demonstrate the completeness of information retention. Relevant experiments have also confirmed that our proposed shifting method effectively preserves information throughout the network. Our operator exhibits significantly higher mutual information than Shift-sa, showcasing its efficacy in information preservation.
>
> **3, real-world.**
>
> Thank you for your feedback. Due to time constraints, we initially conducted experiments on real-world remote sensing satellite scenes, which involve more complex degradations compared to the simulated scenarios with simple down-sampling and blurring. The results were as follows:
>
> - **PANNET**:
>   - D_{\lambda} = 0.0737
>   - D_s = 0.1224
>   - QNR = 0.8143
>
> However, with our newly designed network, the updated metrics are:
>
> - **New Network**:
>   - D_{\lambda} = 0.0716
>   -  D_s = 0.1215
>   - QNR = 0.8156
>
> Here, lower values of D_s and  D_{\lambda}, along with a higher QNR, indicate better performance. These improvements demonstrate that our approach continues to perform effectively in real-world scenarios. We will update the paper to reflect these results and emphasize the effectiveness of our method in practical applications.
>
> **4, comlexity.**
>
> We understand that the paper’s content might not fully align with the title. To address this, we will revise the paper to better demonstrate how our method enhances small-size restoration networks. This revision will include a thorough discussion of its impact on parameters, FLOPs, and inference time, particularly for tasks such as low-light image enhancement using networks like SID. We will ensure that the paper clearly explains the application and evaluation of our proposed concept in this context. We also acknowledge that the implementation details might be unclear. To clarify, we will explicitly define the baseline or original model used for comparison, including a detailed description of the model architecture, configurations, and any modifications. Due to time constraints, we tested our method on the DnCNN model. In Table 5, we compared our Fourier shifting operator with the spatial shift operator and the baseline 3x3 convolution. Our Fourier shifting approach not only halved the number of parameters but also improved network performance, whereas the spatial shift operator caused a significant 14 dB performance drop. Additionally, we assessed runtime and FLOPs to evaluate suitability for edge devices. Our results show that the DnCNN-18 model with Fourier shifting achieved an average runtime of 7 milliseconds and 1.5 GFLOPs per 256x256 image on an NVIDIA GTX 1080 GPU, compared to 15-20 milliseconds and 3.9 GFLOPs for the baseline. This substantial reduction in both runtime and FLOPs highlights the efficiency of our method for resource-constrained hardware.

---

> > ### Comment · Reviewer_3jgW · 2024-08-13
> >
> > Thank you for your response.
> >
> > I appreciate the efforts made by the authors to address the concerns. However I feel that the authors wrote the rebuttal in hurry without citing the mentioned work [PANNET] and comlexity for complexity in 4.
> >
> >  But, I am still worried about the contributions aligned with the title, as discussing these facts would increase the relevance of their proposed work. I still feel that manuscript needs to be polished alot , like the algos mentioned in section 2.1 could be moved into the supp and more details could be incorporated about the important facts like complexity. Additionally, I still feel that the paper is still an extension of the  "When Shift Operation Meets Vision Transformer:An Extremely Simple Alternative to Attention Mechanism" in Fourier domain, thus I would like to keep my score.

---

### Official Review · Reviewer_hJxw · 2024-07-13

**Soundness:** 3
**Presentation:** 3
**Contribution:** 3
**Rating:** 6
**Confidence:** 4

**Summary:**

This paper explores the use of a spatial-shift operator for image restoration tasks, addressing the issue of information loss identified through experimental analysis. The authors propose a shift operator in the Fourier domain, leveraging the periodicity and cycling properties of the Fourier transform to develop a theoretically information-lossless operator that enhances performance. Two variants of the operator, magnitude-phase and real-imaginary, are considered. Experimental results across various image restoration tasks demonstrate superior performance compared to baseline methods and the spatial-shift operator. Interestingly, one could replace multiple spatial shifts with the proposed Fourier-shifts to reduce the number of parameters and thus potentially reduce memory requirements.

**Strengths:**

1) The paper presents a conceptually straightforward and theoretically sound idea.

2) The proposed operator enhances the performance of image restoration tasks. Simultaneously, depending on its design, it could potentially reduce computational complexity.

**Weaknesses:**

The lower portion of Table 5 indicates that the number of parameters remains constant when multiple Fourier-shift operators are substituted with each spatial-shift operator. However, crucial metrics such as runtime and FLOPs are not included. Solely relying on the number of parameters does not provide a comprehensive understanding of the ablation’s suitability for the ultimate objective of deployment on edge or mobile hardware.

**Questions:**

In Figure 6-right, what is the cause of the artifacts present before the application of Shift-sa? Shouldn’t the two feature maps be identical prior to the application of either of the two operators?

Minor issues:

- In Table 4, the ERGAS metric value for the Shift-sa method on the WorldView-II dataset appears to be erroneously formatted in bold.

- Table 5 does not display any gray backgrounds in my PDF viewer. The statement on Line 233, “To further elucidate this point, values falling below the baseline are marked with a gray background, highlighting their relevance and impact,” ([pdf](zotero://open-pdf/library/items/9U2B2N4L?page=9)) may be misplaced.

**Limitations:**

No comments

---

> ### Author Rebuttal · Authors · 2024-08-04
>
> **1, Complexity.**
>
> Thank you for your feedback. In Table 5, we compared the performance of our proposed Fourier shifting operator against the spatial shift operator and the baseline 3x3 convolution. Replacing convolutions with Fourier shifting not only reduced the number of parameters by half but also improved network performance, whereas the spatial shift operator led to a significant 14 dB performance drop. Additionally, we evaluated runtime and FLOPs to assess suitability for edge devices. Our experiments show that the DnCNN-18 model with our Fourier shifting configuration achieves an average runtime of 7 milliseconds and 1.5 GFLOPs per 256x256 image, compared to 15-20 milliseconds and 3.9 GFLOPs for the baseline on an NVIDIA GTX 1080 GPU. This indicates a substantial reduction in both runtime and FLOPs, highlighting the efficiency of our method for deployment on resource-constrained hardware. We will update Table 5 to include these metrics for a comprehensive evaluation. The lower portion of Table 5 uses a controlled variable approach to ensure consistent parameters while evaluating the network's robustness to shifting size. It is important to note that the spatial-shifting configurations were not aligned with our method's parameters throughout. Our results demonstrate that our network exhibits high robustness to varying shifting sizes.
>
> **2, artifacts.**
>
> Thank you for your feedback. Firstly, the Fourier transform is an efficient tool that amplifies image degradations in the Fourier domain, and our learnable parameters act as filters to eliminate these artifacts. Secondly, the artifacts arise from two aspects: (1) features are extracted from low-light degraded inputs, which naturally reflect these degradations, and (2) the testing baseline network inherently uses down-sampling operators to achieve multi-scale features, where the down-sampling inherently causes frequency truncation, leading to frequency aliasing and ringing effects. Additionally, to verify the robustness of our experiments, we analyzed both shallow and deep features in the network, finding that similar artifacts persist and even accumulate, whereas our Fourier shifting can eliminate these, resulting in cleaner features. Our frequency domain approach inherently handles these degradations. The input discrepancies are due to the network being trained from scratch after incorporating our operator, resulting in natural feature changes. Our network effectively reduces degradations both before and after the application of shift-sa, while the degradation persists with shift-sa. Finally, we used mutual information to measure information loss caused by shifting, which validates the effectiveness of our method.
>
> **3,minor issue.**
>
> Thank you for your detailed review and valuable feedback. We will carefully review the entire manuscript to correct any typos and grammar errors to improve the overall clarity and quality of our work.

---

> > ### Comment · Reviewer_hJxw · 2024-08-12
> >
> > I appreciate the authors’ responses to my questions. After reviewing all the feedback, I am particularly concerned about the responses to the issues raised by Reviewer 3jgW. While it is positive that the authors have acknowledged and addressed some of these issues, the necessity of making significant changes to the title suggests that the impact of the work may be less substantial than initially claimed. Consequently, I concur with the reviewer that the contribution may be perceived as more limited. Therefore, I am lowering the score to a weak accept.

---

### Decision · Program_Chairs · 2024-09-25

**Decision:**

Accept (poster)

**Comment:**

All reviewers agree about the originality and the PSNR performance of the proposed solutions.

As pointed out by reviewer 3jgW, this paper did not mention any results regarding the speed or floating point operations. During rebuttal authors have provided results on DNCNN-18, which shows a significant improvement on FLOPS. During the discussion period reviewers discussed that this results should be scalable to other architectures as well.

Re viewer RB41 pointed out to some old references in the experiments. The authors provided comparisons with more recent work during the rebuttal.
Experiments for pan sharpening show results for PANNET (2017) and MutNET (2022), but the citations in text mentions two other works in the field. Authors should fix that in their table/text.

In the discussion period with AC, the reviewers came to the consensus that the paper should be accepted.